# The Challenging Management of Short Bowel Syndrome

**DOI:** 10.3390/diagnostics15121532

**Published:** 2025-06-16

**Authors:** Ismini Kountouri, Afroditi Faseki, Alexandra Panagiotou, Christina Sevva, Ioannis Katsarelas, Dimitrios Chatzinas, Konstantinos Papadopoulos, Vasilis Stergios, Stylianos Mantalovas, Vasileios Alexandros Karakousis, Panagiotis Nachopoulos, Athanasios Polychronidis, Mohammad Husamieh, Christos Gkogkos, Marios Dagher, Panagiota Roulia, Amyntas Giotas, Miltiadis Chandolias, Periklis Dimasis, Dimitra Manolakaki, Isaak Kesisoglou, Nikolaos Gkiatas

**Affiliations:** 1Third Surgical Department, AHEPA University Hospital, Medical Faculty, Aristotle University of Thessaloniki, 1 Kiriakidi Street, 54636 Thessaloniki, Greece; christina.sevva@gmail.com (C.S.); kostaspap1995@hotmail.com (K.P.); vasilis_stergios@hotmail.com (V.S.); steliosmantalobas@yahoo.gr (S.M.); alexanderkarakousis@gmail.com (V.A.K.); mariosdag@gmail.com (M.D.); panagiotar96@gmail.com (P.R.); ikesis@hotmail.com (I.K.); 2Department of General Surgery, General Hospital of Katerini, 60132 Pieria, Greece; afroditifaseki@gmail.com (A.F.); alexandrapanayiotou.larissa98@gmail.com (A.P.); giannis24katsarelas@gmail.com (I.K.); jimmys055@gmail.com (D.C.); panosnacho@gmail.com (P.N.); thanospolychronidis@gmail.com (A.P.); 3bbadi1997@gmail.com (M.H.); miltoshandolias@gmail.com (M.C.); dimasis@yahoo.com (P.D.); dimanolakaki@gmail.com (D.M.); nikgiat71@gmail.com (N.G.); 3Gynecology and Obstetrics Department, General Hospital of Katerini, 60132 Pieria, Greece; akisgogos@yahoo.gr (C.G.); ammag10@live.com (A.G.)

**Keywords:** acute abdomen, mesenteric ischemia, short bowel syndrome, excessive bowel resection, management

## Abstract

A 62-year-old female presented to the Emergency Department of the General Hospital of Katerini, Greece, complaining of abdominal pain, fever, and general discomfort. Laboratory tests indicated an elevated white blood cell count and an elevated C-reactive protein level. A computed tomography (CT) scan revealed dilated small bowel loops and free intraperitoneal fluid. During laparotomy, extensive ischemia and necrosis of both the small and large bowel were discovered, and a resection of the small bowel and the right colon was performed, leaving the patient with only 90 cm of small intestine and a jejunocolic anastomosis. Postoperative management was particularly challenging, requiring a multidisciplinary approach, an intensive care unit stay, reoperations due to anastomotic leaks, continuous parenteral nutrition and electrolyte management, and aggressive antibiotic treatment for persistent bacterial infections. This case report highlights the importance of appropriate management of this life-threatening complication following extensive bowel resection.

**Figure 1 diagnostics-15-01532-f001:**
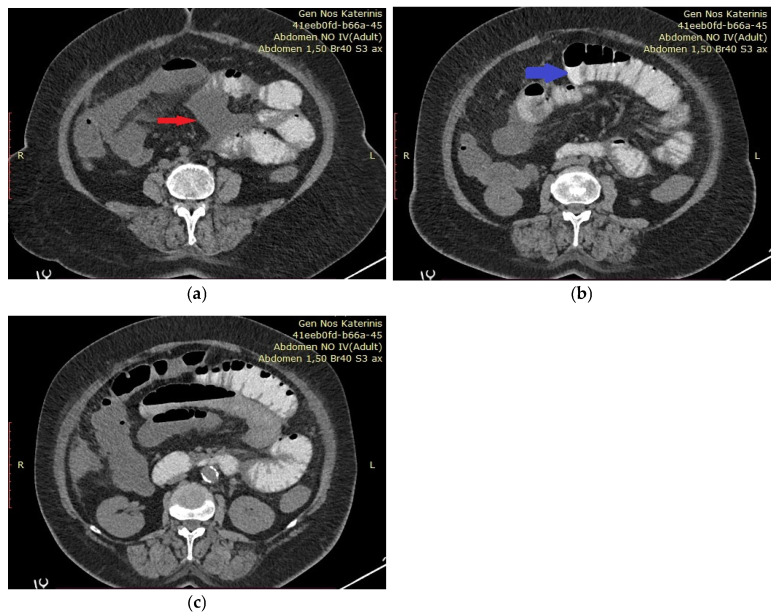
(**a**–**c**) We present the case of a 62-year-old woman who presented to the Emergency Department of the General Hospital of Katerini, Greece, complaining of acute abdominal pain, fever, and general discomfort. She reported that she was suffering from non-Hodgkin lymphoma and was obese. She received no medication and reported no other medical conditions. The patient was initially hemodynamically stable, and physical examination revealed generalized abdominal tenderness upon palpation. Laboratory tests showed an elevated white blood cell (WBC) count of 14.74 × 103/μL and a C-reactive protein (CRP) level of 32 mg/dL, while her hematocrit (HCT) was 28.6% and her hemoglobin was 10.6 g/dL. Due to the patient’s reported allergies, an abdominal computed tomography (CT) scan without intravenous contrast was performed, revealing dilated small bowel loops and free intraperitoneal fluid, without evidence of bowel obstruction or signs of malignancy. (**a**) The red arrow indicates free intraperitoneal fluid. (**b**) Dilated small bowel loops are indicated by the blue arrow. (**c**) The air–fluid levels are indicative of ileus.

**Figure 2 diagnostics-15-01532-f002:**
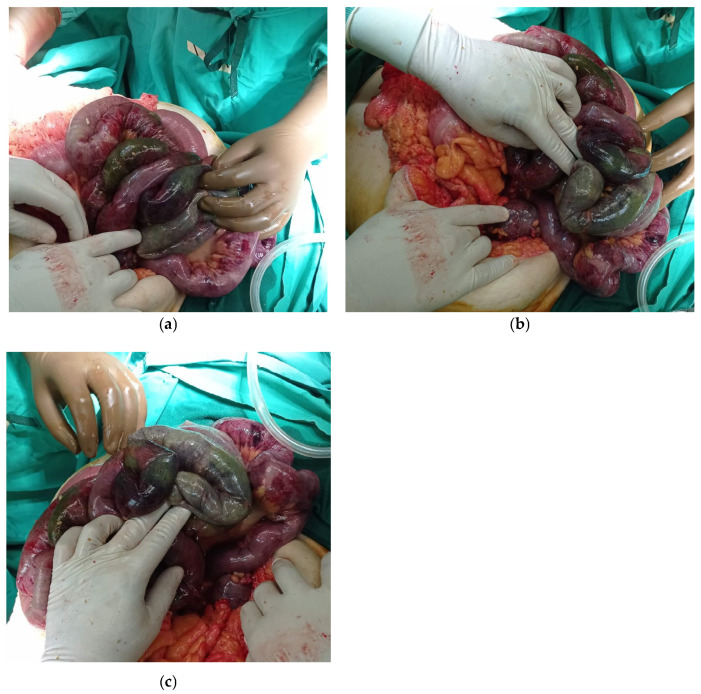
(**a**–**c**) A nasogastric tube was inserted, and a conservative approach was initially adopted. The patient was admitted to the Surgical Clinic but presented with dramatic clinical deterioration three hours later, including abdominal cramping, hemodynamic instability, and persistent fever unresponsive to aggressive analgesic and antipyretic treatment. Due to the patient’s hemodynamic status and the clinical signs of an acute abdomen, the decision was made to proceed with an emergency exploratory laparotomy. During laparotomy, the small and large bowels were found to be necrotic and nonviable, and an extensive enterectomy was performed. The ischemic small bowel and right colon were resected, leaving the patient with only 90 cm of small intestine and a jejunocolic anastomosis. The patient was transferred to the intensive care unit (ICU), intubated, and placed on inotropic support. While in the ICU, she developed persistent diarrhea and electrolyte imbalances, including hypokalemia and hypernatremia, which did not improve despite continuous intravenous administration of electrolytes and fluids. She presented with immunoglobulin deficiency as a result from protein-losing enteropathy. Parenteral nutrition was initiated immediately to support surgical healing, promote ileus resolution, and prevent vitamin and mineral deficiencies. The parenteral regimen included balanced administration of amino acids, intravenous lipids, and glucose-based carbohydrates. A proton pump inhibitor was administered daily to control gastric acid hypersecretion. The patient remained afebrile and hemodynamically stable during her ICU stay. Enteral feeding was initiated on the fifth postoperative day and was high in protein and supplemented with specialized protein-enriched formulas. On the tenth postoperative day, while the patient remained hemodynamically stable and afebrile, bile was observed in the surgical drain, raising suspicion of an anastomotic leak. A second laparotomy was performed, which revealed a small leakage point at the jejunocolic anastomosis. The anastomosis was reconstructed, and the patient was returned to the ICU.She remained in the ICU for two more weeks, during which she was gradually extubated, began enteral feeding, and remained afebrile, hemodynamically stable, and without signs of anastomotic leakage. Parenteral nutrition and proton pump inhibitor administration continued throughout her ICU stay. Diarrhea and electrolyte imbalances persisted even after the patient was transferred back to the surgical clinic.

**Figure 3 diagnostics-15-01532-f003:**
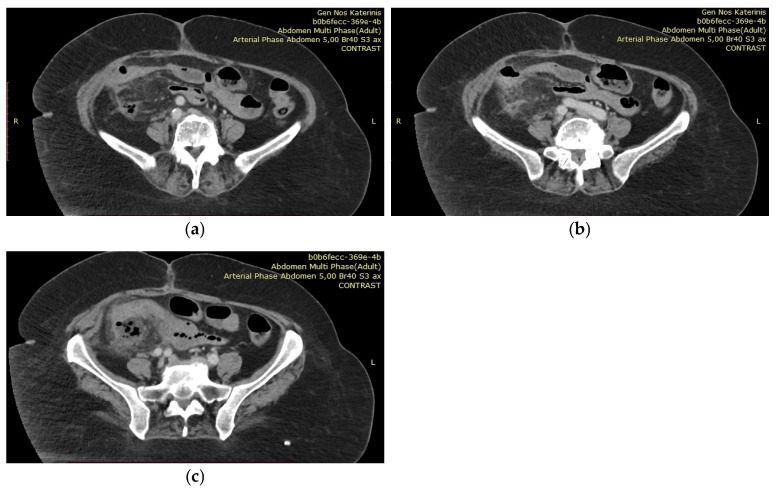
(**a**–**c**) A few days later, she developed a fever of 39 °C. Blood cultures revealed a Pseudomonas infection, which was treated with targeted intravenous antibiotic therapy (ceftolozane/tazobactam). However, as the fever persisted one week into treatment, a new abdominal CT scan was performed. The imaging revealed a recurrent small anastomotic leak and the formation of a perianastomotic abscess. The patient was then transferred to the 3rd Surgical Department of AHEPA University Hospital for CT-guided drainage placement. Following the procedure, she remained in the department for one month. During this time, she continued to experience diarrhea and electrolyte imbalances and was evaluated by multiple specialties: nephrology for fluid and electrolyte management, infectious disease for persistent Pseudomonas infection, hematology for thrombocytopenia, and gastroenterology for short bowel syndrome. Parenteral nutrition was gradually discontinued, and her diet was adjusted under the guidance of a nutritionist. The patient was discharged once the infection had resolved and her electrolyte levels had stabilized, with specific instructions regarding her diet and required supplements. However, diarrhea persisted throughout her hospitalization and after returning home. Due to ongoing weight loss and persistent diarrhea, she was scheduled for Port-a-Cath placement two weeks after discharge in order to receive home parenteral nutrition. Unfortunately, the patient was readmitted to the General Hospital of Katerini after three days of hospital discharge, presenting with general discomfort and disorientation. Her electrolyte imbalance recurred, with potassium at 2.32 mmol/L and sodium at 176 mmol/L. She was hospitalized again, received parenteral nutrition and fluids, and was discharged 10 days later. After three more hospitalizations for similar symptoms and conditions, she eventually passed away at home two months later. With this case report, we aim to highlight the challenges in managing patients with short bowel syndrome and emphasize the importance of a multidisciplinary approach for this clinical condition. Short bowel syndrome is defined as the condition in which less than 180 to 200 cm of small intestine remains, necessitating ongoing nutritional and fluid support [1,2]. Various pathologies can lead to the development of this syndrome, including Crohn’s disease, mesenteric ischemia, radiation enteritis, post-surgical adhesions, and postoperative complications, such as excessive resection in our case [1,2,3]. Approximately 75% of cases result from massive bowel excision, while the remaining 25% are due to multiple resections [1]. Cases involving ileocecal resection, like ours, represent the most severe form of the syndrome and pose significant management challenges [3]. Short bowel syndrome can be classified into three types: jejunostomy, jejunocolic anastomosis, or jejunoileal anastomosis, all of which result in loss of intestinal absorptive surface [1,2,3]. Three phases of this condition have been identified: the acute phase, the adaptation phase, and the maintenance phase [1], each requiring specific management strategies to achieve successful patient rehabilitation. The acute phase begins immediately after resection [3] and lasts 3 to 4 weeks. It is characterized by dehydration with acute kidney failure, acid–base abnormalities, and electrolyte deficiencies, necessitating close monitoring and hospitalization [1]. During this phase, parenteral nutrition is administered to provide nutritional and fluid support [2], while enteral feeding is started as soon as the gastrointestinal tract is able to absorb nutrients [2]. The adaptation phase, lasting 1 to 2 years, is when the remaining bowel adapts to enhance nutrient absorption, slow intestinal transit to maximize absorption time, and undergo adaptive hyperphagia [1,3]. Parenteral nutrition is gradually reduced and replaced by enteral feeding during this phase. The maintenance phase is the final stage and requires special diets, oral or intramuscular nutrients, and pharmacological supplements [1]. At this point, the intestine has reached maximum adaptation, and parenteral nutrition is usually no longer necessary [2]. The main complications of short bowel syndrome arise from malabsorption and include malnutrition, weight loss, steatorrhea, diarrhea, electrolyte imbalances, and vitamin deficiencies [1]. Secondary complications include nephrolithiasis due to hyperoxaluria, cholelithiasis, transient gastric hypersecretion, bacterial overgrowth, dehydration, hyponatremia, potassium and magnesium deficiencies, renal failure, calcium oxalate kidney stones, cholestasis secondary to intestinal failure-associated liver disease, gallstones, and D-lactic acidosis [1]. Patients with short bowel syndrome should be managed individually and often become experts in coping with their condition [3]. Achieving a good quality of life should remain a primary goal when treating these patients [3], while those with irreversible intestinal failure, as in our case, often die prematurely [3] and therefore require more intensive treatment. Treatment strategies should focus on supporting the patient to prevent the syndrome’s complications [1,3]. The primary goal in patients with short bowel syndrome is to maintain adequate nutritional status through early postoperative parenteral nutrition and enteral feeding [1]. Patients require higher energy intake, which can be provided through oral sip feeds or high-energy enteral feeds administered at night via a nasogastric or gastrostomy tube, with gradual reduction or cessation once normal weight is regained [3]. The ideal diet for these patients is high in carbohydrates (polysaccharides), normal (not restricted) in fat (long-chain triglycerides), and low in oxalate [3]. In cases like ours, where intestinal failure is irreversible, more permanent treatment options may be necessary. These include lifelong home parenteral nutrition or intestinal transplantation, especially if a patient absorbs less than one third of oral energy intake, has high energy requirements with absorption between 30 and 60%, or experiences socially unacceptable diarrhea or large stomal output when increasing oral/enteral intake [1,3]. Patients with short bowel syndrome generally have a low quality of life [4] due to constant weight loss and persistent diarrhea [3]. Diarrhea can be managed with loperamide 2–8 mg taken half an hour before meals, and occasionally codeine phosphate (30–60 mg half an hour before meals) is added [3]. Due to the risk of dehydration, hyponatremia, chronic renal failure, and nephrolithiasis, early recognition and management are crucial. The goal is to maintain normal hydration with a daily urine output of 800 mL and a urine sodium level greater than 20 mmol/L [1]. Nutritional evaluation by professional nutritionists is essential in the lifelong management of patients suffering from short bowel syndrome [5]. These specialists assess the patient’s overall condition using specific laboratory tests (e.g., electrolytes, liver and kidney function), as well as monitoring fluid balance, weight changes, serum micronutrient levels, and bone density [5]. In cases of short bowel syndrome, the remaining bowel often adapts by increasing in length and diameter, accompanied by adaptive hyperplasia of the small intestinal mucosa [5], which enhances the absorptive surface area of the remaining intestine [3,5]. Surgical strategies can also be employed to restore intestinal continuity and improve the function of the remaining bowel [2]. For patients with jejunostomy, restoration procedures should be attempted whenever feasible. In cases of extensive bowel resections, surgical interventions may include autologous reconstruction of the gastrointestinal tract, enteroplasties (such as longitudinal intestinal lengthening and tapering (LILT), serial transverse enteroplasty (STEP), or other tapering enteroplasties), or intestinal transplantation [2,6]. To our knowledge, there is currently no evidence linking short bowel syndrome with anastomotic leakage following gastrointestinal surgery, although anastomotic ulcerations have been reported in affected patients [7]. This case report aims to raise awareness among surgeons about the challenges of managing patients with short bowel syndrome after extensive bowel resections. Given the numerous complications associated with this condition, surgeons should exercise caution when extensive resections are necessary and strive to preserve as much bowel as possible. Due to the complexity of this clinical scenario, a multidisciplinary approach is essential in the management of patients with short bowel syndrome.

## Data Availability

No new data were created or analyzed in this study.

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
