# Peer review of "The Challenging Management of Short Bowel Syndrome"

_diagnostics, 2025, doi:10.3390/diagnostics15121532_

Round 1
Reviewer 1 Report
Comments and Suggestions for Authors
This is a nice report that is well-written. There are some clarifications that need to be made before the manuscript is accepted.
These include:
- I was not clear what the exact aetiology for the ischaemic bowel. Was this mesenteric ischaemia presumably from some sort of thrombotic event given the time course?
- The patient passed away at home after 3 more hospitalisations with the same symptoms and condition. Why did she pass away? The same symptoms and condition referred to are general discomfort, disorientation and electrolyte abnormalities. Would this have caused her death through a cardiac event secondary to severe electrolyte disturbance? Or was it more likely through sepsis? Was she palliated at home?
- Can the authors comment on why there was recurrent anastomotic leakage? Was the bowel itself extremely friable and fragile at the time of surgery?
There are several minor English language corrections that should be done:
- On page 4 out of 7 the authors write "platelet count drop". It would be best to use the term thrombocytopenia
- The following sentence is long with several 'ifs' and should be reworded. "In cases like ours, where the intestinal failure is irreversible, if a patient absorbs less than one third of the oral energy intake, if there are high energy requirements and absorption is 30–60%, or if increasing oral/enteral nutrient intake causes a socially unacceptably amount of diarrhea or a large volume of stomal output, a more permanent treatment option might be necessary, requiring life long home parenteral nutrition or intestinal transplantation [1], [3].
Author Response
Comment 1 I was not clear what the exact aetiology for the ischaemic bowel. Was this mesenteric ischaemia presumably from some sort of thrombotic event given the time course?
Responce 1 : The patient was suffering from lymphoma and thus was in a thrombophylic situation. The ischaemia was persumed to be a result of a thombotic event
Comment2:The patient passed away at home after 3 more hospitalisations with the same symptoms and condition. Why did she pass away? The same symptoms and conditions referred to are general discomfort, disorientation and electrolyte abnormalities. Would this have caused her death through a cardiac event secondary to severe electrolyte disturbance? Or was it more likely through sepsis? Was she palliated at home?
Response 2: Unfortunately, after the continuous hospitalisations, the patient lost all will to live. She refused to visit the ER and died at her home. She was under psychiatric observation and was receiving treatment
Comment 3: Can the authors comment on why there was recurrent anastomotic leakage? Was the bowel itself extremely friable and fragile at the time of surgery?
Response 3: As it is well known, anastomotic leaks can be a result of systemic factors, such as malnutrition, parenteral nutrition, and underlying factors like kidney failure and obesity. Our patient had all of these factors, and she was also in a thrombophilic situation.
Comment 4
- On page 4 out of 7 the authors write "platelet count drop". It would be best to use the term thrombocytopenia
- The following sentence is long with several 'ifs' and should be reworded. "In cases like ours, where the intestinal failure is irreversible, if a patient absorbs less than one third of the oral energy intake, if there are high energy requirements and absorption is 30–60%, or if increasing oral/enteral nutrient intake causes a socially unacceptably amount of diarrhea or a large volume of stomal output, a more permanent treatment option might be necessary, requiring life long home parenteral nutrition or intestinal transplantation [1], [3].
Response 4: We corrected accordingly. Also, excessive English editing was applied to the manuscript.
Reviewer 2 Report
Comments and Suggestions for Authors
The authors publish a sad and instructive case report about a 62-year-old female patient suffering from extensive intestinal necrosis, whose life could only be temporarily saved at the cost of short bowel syndrome.
Although the case is interesting, the description of the underlying diseases and the image documentation are incomplete in several respects.
I have the following comments on the material:
- Why is the article included in the “interesting images” category? The native CT images do not shed light on a number of issues.
- What illnesses did the 62-year-old woman have? What was the cause of the extensive intestinal necrosis? When did her symptoms begin?
- Did she have atherosclerosis, acquired or genetic thrombophilia? Did she have a metabolic disorder?
- What medications was she taking? Were any of them likely to predispose her to this condition?
- Did she have heart disease or arrhythmia?
- Where were the recurrent Pseudomonas infections located? How were they treated?
- What were the patient's immunoglobulin levels? Was there any evidence of protein-losing enteropathy?
- What exactly was the patient's parenteral and enteral nutrition?
- Did the patient receive teduglutide treatment for short bowel syndrome?
From the above, it is clear that although the case is interesting, the case presentation falls short of expectations in a number of respects.
The article is not suitable for publication in its current form.
Author Response
Comment 1
Why is the article included in the “interesting images” category? The native CT images do not shed light on a number of issues.
Response 1 : The intraoperative images of the excessive necrosis and the CT scans with the recurrent anastomotic leaks are interesting images in our opinion.
Comment 2: What illnesses did the 62-year-old woman have? What was the cause of the extensive intestinal necrosis? When did her symptoms begin?
Did she have atherosclerosis, acquired or genetic thrombophilia? Did she have a metabolic disorder?What medications was she taking? Were any of them likely to predispose her to this condition. Did she have heart disease or arrhythmia?
Response 2: The patient was suffering from Non - Hodgkin lymphoma and obesity, both thrombophilic situations. We added the information in the text. The symptoms began one hour before arriving in the ER. We added the word acute in the text. She reported no other medical conditions and was receiving no medications.
-Comment 3: Where were the recurrent Pseudomonas infections located? How were they treated?
Response 3: She had a blood Pseudomonas infection , as reported in the text and was treated with ceftolozane/tazobactam
Comment 4: What were the patient's immunoglobulin levels? Was there any evidence of protein-losing enteropathy?
Response 4: Her immunoglobulin levels post-operatively were continuously low, and thus the diagnosis of a protein-losing enteropathy due to the excessive resection was made. We added the information on the text.
Comment 5: What exactly was the patient's parenteral and enteral nutrition?
Response 5: The parenteral regimen included balanced administration of amino acids, intravenous lipids, and glucose-based carbohydrates. Enteral nutrition was high in protein and supplemented with specialized protein-enriched formulas.
Comment 6: Did the patient receive teduglutide treatment for short bowel syndrome?
Response 6: Since the patient was able to eat enteral nutrition and required no parenteral nutricion, teduglutide treatment was not administrated.
Round 2
Reviewer 2 Report
Comments and Suggestions for Authors
I would like to thank the authors for accepting and adopting my suggestions. Even if a patient is admitted to the ER as an emergency, when describing a case, we must strive to provide as complete a picture of the patient as possible, including their medical history and any circumstances that may help the reader better understand why severe intestinal necrosis may have developed. The revised version of the article is now acceptable for publication.